# Age-Related Structural and Functional Changes of the Hippocampus and the Relationship with Inhibitory Control

**DOI:** 10.3390/brainsci10121013

**Published:** 2020-12-19

**Authors:** Sien Hu, Chiang-shan R. Li

**Affiliations:** 1Department of Psychology, State University of New York at Oswego, Oswego, NY 13126, USA; 2Department of Psychiatry, Yale University School of Medicine, New Haven, CT 06519, USA; chiang-shan.li@yale.edu; 3Department of Neuroscience, Yale University School of Medicine, New Haven, CT 06520, USA; 4Interdepartmental Neuroscience Program, Yale University School of Medicine, New Haven, CT 06520, USA

**Keywords:** age, hippocampus, inhibition

## Abstract

Aging is associated with structural and functional changes in the hippocampus, and hippocampal dysfunction represents a risk marker of Alzheimer’s disease. Previously, we demonstrated age-related changes in reactive and proactive control in the stop signal task, each quantified by the stop signal reaction time (SSRT) and sequential effect computed as the correlation between the estimated stop signal probability and go trial reaction time. Age was positively correlated with the SSRT, but not with the sequential effect. Here, we explored hippocampal gray matter volume (GMV) and activation to response inhibition and to p(Stop) in healthy adults 18 to 72 years of age. The results showed age-related reduction of right anterior hippocampal activation during stop success vs. go trials, and the hippocampal activities correlated negatively with the SSRT. In contrast, the right posterior hippocampus showed higher age-related responses to p(Stop), but the activities did not correlate with the sequential effect. Further, we observed diminished GMVs of the anterior and posterior hippocampus. However, the GMVs were not related to behavioral performance or regional activities. Together, these findings suggest that hippocampal GMVs and regional activities represent distinct neural markers of cognitive aging, and distinguish the roles of the anterior and posterior hippocampus in age-related changes in cognitive control.

## 1. Introduction

Memory impairment is the most important characteristic of age-related cognitive deficits, as observed in individuals with mild cognitive impairment (MCI) and Alzheimer’s Disease (AD). The processing of contextual information depends critically on the hippocampus [1,2]. Indeed, hippocampal atrophy represents a biomarker of MCI and early-stage AD [3,4]. Along with impairment in spatial [5] and verbal memory [6] task performance, hippocampal atrophy is associated with higher risk of developing AD. Studies have also reported reduced hippocampal activities [7] and connectivities [8,9], as well as increased hippocampal metabolism [9], in MCI. In particular, hippocampal structural and functional changes may predict cognitive decline [10,11]. Together, hippocampal dysfunction represents a critical etiological marker of age-related cognitive decline and AD.

Inhibitory control, a component function of cognitive control, involves stopping the current or pre-potentiated actions, and allows flexibility in behavioral control in response to conflicting situations. Impairments in inhibitory control has been reported in patients with MCI [12] and early-stage AD [13]. Patients with MCI appeared to exhibit disproportionate decline in inhibitory control among the different domains of executive functions [14]. Compared with age-matched healthy adults, individuals with MCI showed inhibitory deficits as reflected in longer stop signal reaction time (SSRT) in the stop signal task [15], longer reaction time (RT) cost in incongruent conditions in the flanker task [16], lower hit rates during anti-saccades [17], higher interference effect in the Stroop task [18], higher error rate in the Simon task [19], and increased false alarms in the go/no-go task [20]. These studies suggested that inhibitory control may represent a determinative factor of age-related cognitive decline, and should be included in neuropsychological assessments of age-related cognitive dysfunction [14].

The hippocampus can be segmented in the long axis to anterior and posterior (ventral and dorsal, respectively, in rodents) sub-regions, with both shared and distinct anatomical connectivity [21,22,23,24]. The anterior hippocampus receives inputs from the amygdala, hypothalamus, and insular and ventromedial prefrontal areas, while the posterior hippocampus receives inputs from the cingulate cortex, visual cortices, dorsolateral prefrontal cortex, medial temporal, and inferior parietal lobes [24,25]. Connectivity studies revealed that the anterior hippocampus is functionally connected to the limbic system, and the posterior hippocampus to the inferior frontal regions and parietal-temporal junction [26,27]. Other studies distinguished the functions of the hippocampal sub-regions, with the anterior hippocampus involved in affective and motivational processing [28], contextual memory [29], imagination, and memory encoding [30], and the posterior hippocampus in spatial processing [31], local representation [32], and memory retrieval [27]. On the other hand, human imaging studies have largely not specifically distinguished the anterior and posterior hippocampus in characterizing age-related structural and functional changes.

The hippocampus supports cognition not only in the domains of episodic and verbal memory, but also in processing speed and executive control [33]. Atrophy of the hippocampus was considered a culprit of age-related cognitive decline [34]. In neurotypical adults, reduced hippocampal volume was associated with lower fluid intelligence in the elderly, but not in the young [35]. Smaller hippocampus volume was associated with lower processing speed, as measured by reaction times, in elderly people performing various cognitive tasks [36]. The hippocampus also supports cognition through its anatomical connection to the executive control network, including the prefrontal cortex [37]. Positive hippocampal–prefrontal cortical connectivity was observed in young and healthy older adults during retrieval of face–name paired association memory, but not in older adults with signs of pathological aging and minor neurocognitive disorder [38]. Nonetheless, very few human studies have directly examined hippocampal contribution to inhibitory control.

On the other hand, a few animal studies have highlighted the roles of the hippocampus in inhibitory control. For example, rats with lesions in the ventral hippocampus were unable to inhibit impulsive responses to wrongful locations and during wrongful time periods in a choice task [39,40]. A recent study reported that the neuropeptide galanin, which binds to G-protein-coupled receptors densely expressed in the ventral hippocampus, plays a central role in impulse control [41]. In an experiment where rats learned to associate an object and place for rewards, hippocampal neurons showed higher activity during inhibition of responses to wrongful objects and places [42]. In a visual discrimination task with radial maze, rats with a lesioned or intact hippocampus showed no difference in context-specific learning; however, when the context was reversed (i.e., reinforced arms become unreinforced), rats with a lesioned relative to those with an intact hippocampus showed less difficulty in reversal learning, suggesting that the acquired inhibition to the unreinforced arms in the original context was stored in the hippocampus [43]. Further examination revealed that rats with ventral but not those with dorsal hippocampal lesions showed impaired inhibition towards unreinforced arms, suggesting a specific role of the ventral hippocampus in acquiring inhibitory associations [44]. These studies together suggest the potential importance of the hippocampus in inhibitory control.

The goal of the current study is to examine age-related changes in hippocampal gray matter volumes and hippocampal activation during inhibitory control. In our previous studies, we have shown that age is associated with impaired reactive inhibitory control and relatively intact proactive control in the stop signal task [45,46]. Here, targeting the hippocampus, we (1) examined hippocampal activation during reactive inhibitory and proactive control in older adults, and (2) explored whether the anterior and posterior hippocampus may partake in these age-related processes differently.

## 2. Materials and Methods

### 2.1. Participants

We examined the same data set of 149 adults (83 women) 18 to 72 (31.6 ± 11.9; mean ± SD) years of age [45,46]. Age and sex frequencies are shown in Figure A1. All participants were physically healthy, with no major medical illnesses or current use of prescription medications. None of them reported having a history of head injury or neurological or psychiatric illness. All participants signed a written consent after they were given a detailed explanation of the study in accordance with a protocol approved by the Yale Human Investigation Committee [45,46].

### 2.2. Stop Signal Task and Stop Signal Reaction Time

Participants performed a standard stop signal task (SST), in which a circle—the “go” signal—prompted participants to press a button quickly, and a cross—the “stop” signal—following the go signal instructed participants to withhold the button press [47,48,49]. Go (~75%) and stop (~25%) trials were randomized in presentation. The time between the go and stop signals, the stop signal delay (SSD), started at 200 ms and varied from one stop trial to the next according to a staircase procedure, increasing and decreasing by 67 ms each after a successful and failed stop trial [50]. Participants were trained briefly on the task prior to the imaging session. They were instructed to press the button quickly when they saw the go signal while keeping in mind that a stop signal might come up in some trials. In the scanner, 146 participants completed four 10-min sessions of the task, and three completed three sessions, with approximately 100 trials in each session [46]. Hence, 146 participants completed a total of approximately 300 go and 100 stop trials, and three completed approximately 225 go and 75 stop trials. The stop signal reaction time (SSRT) was computed for each participant, and a longer SSRT suggested lesser capacity of inhibitory control [51].

### 2.3. A Bayesian Model of Proactive Control

As in our previous work [45,52,53], we used a dynamic Bayesian model [54] to estimate the prior belief of an impending stop signal on each trial based on prior stimulus history. In the model, subjects believe that stop signal frequency *r_k_* on trial *k* has a probability α of being the same as *r_k−_*_1_, and probability (1 − α) of being re-sampled from a prior distribution π(*r_k_*). Subjects are also assumed to believe that trial *k* has a probability *r_k_* of being a stop trial, and probability 1 − *r_k_* of being a go trial. With these generative assumptions, subjects used Bayesian inference to update their prior belief of seeing a stop signal on trial *k*, *p*(*r_k_*|*S_k−_*_1_), based on the prior or the last trial *p*(*r_k−_*_1_|*S_k−_*_1_) and last trial’s true category (*s_k_* = 1 for stop trial, *s_k_* = 0 for go trial), where *S*_k_ [*s*_1_, . . . , *s*_k_] is short-hand for all trials 1 through *k*. Specifically, given that the posterior distribution was *p*(*r_k−_*_1_|*S_k−_*_1_) on trial *k −* 1, the prior distribution of stop signals in trial *k* is given by:*p*(*r_k_*|*S_k−_*_1_) = α *p*(*r_k−_*_1_|*S_k−_*_1_) + (1 − α) π(*r_k_*),(1)
where the prior distribution π(*r_k_*) is a beta distribution with prior mean pm and shape parameter scale, and the posterior distribution is computed from the prior distribution and the outcome according to the Bayes’ rule:*p*(*r_k_*|*S_k_*)∝*P*(*s_k_*|*r_k_*) *p*(*r_k_*|*S*_*k*−1_).(2)

The Bayesian estimate of the probability of trial *k* being stop trial, which we colloquially call p(Stop) in this paper, given the predictive distribution *p*(*r_k_*|*S_k−_*_1_), is expressed by:(3)$$\begin{array}{cc} P\left(s_{k}=1|S_{k-1}\right) & =\int^{\;}P\left(s_{k}=1|r_{k}\right)P\left(r_{k}|S_{k-1}\right)dr_{k}=\int^{\;}r_{k}P\left(r_{k}|S_{k-1}\right)dr_{k}\\ & =r_{k}|S_{k-1} \end{array}$$

In other words, p(Stop) or the probability of a trial *k* being a stop trial is simply the mean of the predictive distribution *p*(*r_k_*|*S_k−_*_1_). The assumption that the predictive distribution is a mixture of the previous posterior distributions and a generic prior distribution is essentially equivalent to using a causal, exponential, and linear filter to estimate the current rate of stop trials [55]. In summary, for each subject, given a sequence of observed go/stop trials and the three model parameters {α, pm, scale}, we estimated p(Stop) for each trial [45,52,53].

We followed our earlier work in specifying the parameters for Bayesian models [45,52,53]. Specifically, we assumed a prior β distribution, β (3.5, 7.5), equivalent to a prior mean = 0.25, scale = 10, and a learning parameter α = 0.8 for all participants. The mean of the prior distribution was set at 0.25 to reflect the frequency of stop trials. Although an individual participant might present a different optimal set of parameters, individual model parameter estimates tended to be noisy. We followed the standard of model-based fMRI analyses by keeping a fixed set of parameters across the group in characterizing behavior related to stop signal anticipation and regional responses to p(Stop) [52,56,57]. The proactive inhibition index was formulated via the sequential effect as:Seq.Effect = Corr(p(Stop), goRT) (4)

Our previous work showed that the sequential effect was not sensitive to the exact parametrization of the model; a significant correlation between p(Stop) and goRT could be obtained for individual subjects for a wide range of parameters (*r*’s > 0.92; Pearson regression) [52]. The validity of the model was confirmed in a more recent work [53].

### 2.4. MRI Protocol

The image acquisition process was reported in our previous work [45,46,47,48,49,50,53]. Conventional T1-weighted spin-echo sagittal anatomical images were acquired for slice localization using a 3-Tesla scanner (Siemens Trio, Erlangen, Germany). Anatomical images of the functional slice locations were obtained with spin-echo imaging in the axial plan parallel to the Anterior Commissure-Posterior Commissure (AC-PC) line, with repetition time (TR) = 300 ms, echo time (TE) = 2.5 ms, bandwidth = 300 Hz/pixel, flip angle = 60°, field of view = 220 × 220 mm, matrix = 256 × 256, 32 slices with slice thickness = 4 mm, and no gap. A single high-resolution T1-weighted gradient-echo scan was obtained. One hundred and seventy-six slices parallel to the AC-PC line covering the whole brain were acquired with TR = 2530 ms, TE = 3.66 ms, bandwidth = 181 Hz/pixel, flip angle = 7°, field of view = 256 × 256 mm, matrix = 256 × 256, and 1 mm^3^ isotropic voxels. Functional blood oxygenation level dependent (BOLD) signals were then acquired with a single-shot gradient-echo echo-planar imaging (EPI) sequence. Thirty-two axial slices parallel to the AC-PC line covering the whole brain were acquired with TR = 2000 ms, TE = 25 ms, bandwidth = 2004 Hz/pixel, flip angle = 85°, field of view = 220 × 220 mm, matrix = 64 × 64, 32 slices with slice thickness = 4 mm, and no gap. There were three hundred images in each session.

### 2.5. Gray Matter Volumes Derived with Voxel-Based Morphometry (VBM)

Data were analyzed with Statistical Parametric Mapping (SPM12, Wellcome Department of Imaging Neuroscience, University College London, UK). We used the VBM8 toolbox as implemented in SPM to identify differences in the local composition of brain tissues, while discounting large scale differences in gross anatomy and position [46]. T1-images were first co-registered to the Montreal Neurological Institute (MNI) template space using a multiple stage affine transformation. Co-registration started with a coarse affine registration using mean square differences, followed by a fine affine registration using mutual information. Coefficients of the basis functions that minimize the residual square difference (between individual image and the template) were estimated. After affine transformation, T1-images were corrected for intensity bias field and a local means de-noising filter was applied to account for intensity variations (inhomogeneity) and noise caused by different positions of cranial structures within the MRI coil. Images were then segmented into cerebrospinal fluid and gray and white matters using an adaptive maximum a posteriori method with k-means initializations, generating tissue class maps. Segmented and initially registered tissue class maps were normalized, and the normalized gray matter (GM) maps were modulated to obtain the absolute volume of GM tissue corrected for individual brain sizes. Finally, the GM maps were smoothed by convolving with an isotropic Gaussian kernel of 8 mm at full width at half maximum (FWHM) [46].

### 2.6. Preprocessing and Modeling of BOLD Data of the SST

We followed the standard pre-processing of BOLD data as in our previous work [45,46,47,48,49,50,53]. In the pre-processing of BOLD data, images of each participant were realigned (motion-corrected) and corrected for slice timing. A mean functional image volume was constructed for each participant per run from the realigned image volumes. These mean images were co-registered with the high-resolution structural image and then segmented for normalization to an MNI EPI template with affine registration, followed by nonlinear transformation [58,59]. The normalization parameters determined for the structure volume were then applied to the corresponding functional image volumes for each participant. Finally, images were smoothed with a Gaussian kernel of 8 mm at FWHM. Images from the first five TRs at the beginning of each session were discarded, so only signals with steady-state equilibrium between radio frequency pulsing and relaxation were included in data analyses [45,46,47,48,49,50,53].

We employed two generalized linear models (GLM) with four trial outcomes, go success (GS), go error (GE), stop success (SS), and stop error (SE), distinguished for each model [46,47,48,49]. In the first GLM, we modeled BOLD signals by convolving the onsets of the go signals of each trial with a canonical hemodynamic response function (HRF) and the temporal derivative of the canonical HRF [50,59]. We included the reaction time (RT) of GS trials, SSD of SS trials, and SSD of SE trials as parametric modulators in the model, in that order. In the second GLM, we modeled BOLD signals by convolving the onsets of the fixation point (the beginning) of each trial with a canonical HRF and the temporal derivative of the canonical HRF. We included p(Stop) of GS trials, SSD of SS trials, p(Stop) of SS trials, SSD of SE trials, and p(Stop) of SE trials as parametric modulators in the model, in that order. In both models, realignment parameters in all six dimensions were entered in the model. Serial autocorrelation of the time series was corrected by a first degree autoregressive or AR(1) model [60,61]. The data were high-pass filtered (1/128 Hz cutoff) to remove low-frequency signal drifts.

In first-level analysis, we computed a contrast of “stop success > go success” (SS > GS) in the first GLM, and a contrast “1” on the parametric modulator p(Stop) of GS trials in the second GLM for each individual. In second-level analysis, we performed whole-brain regressions of SS > GS against age and p(Stop) > 0 against age, respectively, with small volume correction for the bilateral hippocampus mask obtained from the Automatic Anatomic Labelling (AAL) atlas. Following current reporting standards, all results were examined for voxels meeting a threshold of voxel *p* < 0.05 and corrected for familywise error (FWE) of multiple comparisons on the basis of Gaussian Random Field theory, as implemented in the SPM.

### 2.7. Mediation Analysis

We performed mediation analyses to test the hypothesis that GMV and activations of the hippocampus each mediated the association between age and SSRT using the toolbox developed by Wager and Lindquist (http://www.columbia.edu/cu/psychology/ tor/). The methods were detailed in our previous work [46,49,62]. Briefly, in a mediation analysis, the relation between the independent variable X and dependent variable Y, i.e., X → Y, is tested to see if it is significantly mediated by a variable M. The mediation test is performed by employing three regression equations: Y = i1 + *c*X + *e*1,(5)
Y = i2 + *c*′ X + *b*M + *e*2,(6)
M = i3 + *a*X + *e*3,(7) where i1, i2, and i3 are intercepts, *e*1, *e*2, and *e*3 are residuals, *a* represents X → M, b represents M → Y (controlling for X), *c*′ represents X → Y (controlling for M), and *c* represents X → Y. In the literature, *a*, *b*, *c*, and *c*′ were referred to as path coefficients or simply paths, and we followed this notation. Variable M is said to be a mediator of connection X → Y, if (*c* − *c*′), which is mathematically equivalent to the product of the paths *a × b*, and is significantly different from zero [63]. If the product of *a × b* and the paths *a* and *b* are significant, one concludes that X → Y is mediated by M. In addition, if path *c*′ is not significant, it indicates that there is no direct connection from X to Y and that X → Y is completely mediated by M. Note that path *b* represents M → Y, controlling for X, and should not be confused with the correlation coefficient between Y and M. Note also that a significant correlation between X and Y and between X and M is required for one to perform the mediation test [46,49,62].

## 3. Results

### 3.1. Behavioral Performance

As reported earlier [46], participants responded in 98% ± 3% (mean ± SD) of go trials and 49% ± 3% of stop trials, with 628 ± 119 ms in median go trial reaction time (goRT) and 206 ± 37 ms in SSRT. Both goRT and SSRT were positively correlated with age (*r**_goRT_* = 0.1671, *p* = 0.0416; *r**_SSRT_* = 0.2671, *p* = 0.0010) in linear regressions [46]. On the other hand, the magnitude of sequential effect was not significantly correlated with age (*r* = −0.0510, *p* = 0.5366) [45]. These findings suggested impaired reactive inhibitory control and preserved proactive inhibitory control in older adults.

### 3.2. Age-Related Decreases in Hippocampal Gray Matter Volume (GMV)

With small volume correction (SVC) for the bilateral AAL hippocampus masks, we observed age-related decreases in bilateral hippocampal GMV (Figure 1a). We extracted the GMV of anterior and posterior hippocampus, as defined in Zeidman and Maguire [30] (Figure 1b). The GMVs of anterior and posterior hippocampus were both negatively correlated with age (*r_anterior_* = −0.2554, *p* = 0.0017; *r_posterior_* = −0.2486, *p* = 0.0022), and a slope test [64] revealed no difference between the two regressions (*t* = 0.0446, *p* = 0.9645). We examined whether the GMVs correlated with the SSRT and sequential effect and evaluated the results of Pearson regression at a corrected *p*-value of 0.05/4 = 0.0125. The GMV of the anterior hippocampus was not correlated with SSRT (*r* = −0.1478, *p* = 0.0720) or with the sequential effect (*r* = 0.0653, *p* = 0.4288). The GMV of the posterior hippocampus was not correlated with SSRT (*r* = −0.1541, *p* = 0.0606), or with the sequential effect (*r* = 0.1686, *p* = 0.0398) at the corrected threshold. Table A1 summarizes the statistics of Pearson regressions for the left- and right- hemispheric anterior and posterior hippocampus examined separately.

### 3.3. Age and Hippocampal Activation in Reactive and Proactive Inhibitory Control

With SVC for the bilateral AAL hippocampus masks, we observed a negative age correlation of the right anterior hippocampus (x = 36, y = −22, z = −8, k = 57, Z = 3.31, *p*_FWE_ = 0.033, Figure 2a, blue) activation for SS > GS, while no voxels in the left hippocampus showed significant age-related activations. We also observed a positive age correlation of the right posterior hippocampus (x = 27, y = −40, z = −2, k = 48, Z = 3.67, *p*_FWE_ = 0.011, Figure 2a, red) activation for p(Stop) > 0, while activation of the left hippocampus was not significant (x = −21, y = −40, z = 1, k = 42, Z = 3.00, *p*_FWE_ = 0.077). We visualized these correlations with scatter plots showing the β estimates vs. age. These clusters are also highlighted with an overlay on the anterior and posterior hippocampus masks, as defined in Zeidman and Maguire [30] (Figure 2b,c).

In a separate model, we included sex as a covariate, and found age-related activations peaked in the same coordinates for the anterior hippocampus (Z = 3.51, k = 62, *p*_FWE_ = 0.035) for SS > GS and posterior hippocampus (Z = 3.52, k = 46, *p*_FWE_ = 0.034) for p(Stop) > 0. Sex was also included as a covariate in the regression between age and the β contrast of the anterior and posterior hippocampus, with or without behavioral measures, and sex was not a significant covariate in the regressions (Table A2).

In addition, the β contrast (SS > GS) of the right anterior hippocampus was negatively correlated with SSRT (*r* = −0.2272, *p* = 0.0053), and the β contrast (p(Stop) > 0) of the right posterior hippocampus was not correlated with the sequential effect (*r* = −0.1368, *p* = 0.0963). In the mediation analysis, the β contrast (SS > GS) of the right anterior hippocampus did not mediate the relationship between age and SSRT (Figure 3a).

### 3.4. Age-Related Structural and Functional Changes in the Hippocampus

We performed mediation analyses to examine the relationship between age-related hippocampal structural and functional changes, focusing on voxels in the right anterior hippocampus that overlapped between the regression of GMV with age (Figure 1c green) and the regression of SS > GS with age (Figure 1c blue), and the voxels in the right posterior hippocampus that overlapped between the regression of GMV with age (Figure 1c green) and the regression of p(Stop) > 0 with age (Figure 1c red). The results showed that the GMV of the right anterior hippocampus did not significantly mediate the correlation between age and activity during SS > GS (Figure 3b). The GMV of the right posterior hippocampus did not mediate the relationship between age and activation to p(Stop) (Figure 3c).

## 4. Discussion

Whereas the GMVs of the anterior and posterior hippocampus decreased with age, the GMVs were not correlated with SSRT or with the sequential effect. The activation of the right anterior hippocampus during reactive inhibitory control was negatively associated with age, and the activation of the right posterior hippocampus in proactive control was positively associated with age. These findings suggest potentially distinct roles of the anterior and posterior hippocampus and implicate hippocampal dysfunction in age-related decline in cognitive control.

### 4.1. The Hippocampus and Inhibitory Control

Although rarely a focus in studies of inhibitory control, the hippocampus may partake in cognitive control and adaptive behavior, likely via its connection with the thalamus [65] and prefrontal cortex [66]. In fact, some studies suggested a broader role of the hippocampus, for instance, in volitional finger movements even in the absence of motor learning or recall [67]. An electrophysiological study identified an event-related potential in the hippocampus, likely reflecting task-specific preparation or proactive control for hard mental calculation [68]. A meta-analysis using Neurosynth to identify regions that are recruited in studies of cognitive control identified frontoparietal structures, thalamus, caudate, and hippocampus [69]. Thus, the current findings provide additional evidence in support of hippocampal function in cognitive control.

In particular, we distinguished the potential roles of the anterior and posterior hippocampus, each figuring more prominently in reactive and proactive control. Although no other studies have specifically investigated these roles of the anterior and posterior hippocampus in the stop signal, go/no-go, Stroop, flankers, or other cognitive control paradigms, a few provided evidence in accord with the current findings. For instance, the activation of the hippocampus was reported for the contrast of successful stop > go in a stop signal task, and hippocampal activity was negatively correlated with SSRT [70]. In patients with post-traumatic stress disorder, van Rooij et al. [71] demonstrated higher hippocampal activation during no-go than go trials in a go/no-go task; moreover, this inhibition-related hippocampal activation was negatively correlated with posttraumatic stress scores, suggesting that the importance of the hippocampus in inhibitory control may extend beyond simple cognitive challenges. In a recent work, Meyers et al. [72] examined ventral hippocampal activity during inhibition of threat responses in mice and humans. In humans, participants were conditioned on threatening, safe, and a compound (combination of both threating and safe) signals in the acquisition period. During testing, the ventral hippocampus showed greater activation to the compound than to the safe and threatening signals alone. The authors proposed that the inhibition of threat response was induced by the presentation of a safe signal in the compound condition, and that the ventral hippocampus contributed to such inhibition. Engaged in a similar task, mice showed lower activity in the ventral hippocampus during exposure to threat than to compound and safe signals, suggesting a role of ventral hippocampal neurons in safety monitoring and inhibition of threat response.

### 4.2. A Broader Role of Hippocampal Structure and Function in Age-Related Cognitive Decline

Hippocampal atrophy is a biomarker for MCI and progression to AD [3,73,74]. Reduced hippocampal activations [7], decreased hippocampal resting state connectivity [8,9], and increased hippocampal metabolism [9] were reported in MCI. In particular, hippocampal structural changes and alterations in hippocampal cortical connectivity predicted progression in cognitive decline in MCI [10,11]. The hippocampal volume especially of the right hemisphere was positively associated with the performance in a spatial contextual cueing task in MCI [5]. In encoding faces-to-names, MCI patients showed greater (especially right-hemispheric) hippocampal activation than age-matched healthy controls [75]. In healthy old adults, increased hippocampal activation was associated with increased beta-amyloid accumulation, which predicted longitudinal memory decline [76]. In a longitudinal study of healthy old adults, hyper-activation of the hippocampus at baseline was positively associated with the severity of cognitive decline in two years, supporting hippocampal activation to memory encoding as a predictor of future cognitive impairment [77]. Hippocampus activation exhibited a non-linear trajectory from MCI to AD in that, compared with controls, early phase MCI showed hyper-hippocampal activation and later phase MCI and AD showed hypo-hippocampal activation in the face−name association task [78]. Our results underscoring the roles of the hippocampus in inhibitory control extend this literature in new directions.

Many other studies documented age-related changes in hippocampal function, including failure in recruiting hippocampal connectivity with lateral prefrontal cortical regions during correct rejections of recombined pairs in a stimulus association task [79]. Aging compromised hippocampal-striatal coupling required for integration of episodic information with future value computation in a variant delay discounting task [80]. Both aging and MCI were associated with heightened hippocampal striatal connectivity, along with stronger activation of striatal areas during associative memory retrieval, suggesting functional dedifferentiation [81]. Collected with distinct paradigms, these data together suggest the possibility of both age-related decline and functional compensation in the hippocampal processes to support cognition. Here, we demonstrated that, playing a more significant role in reactive inhibition, the right anterior hippocampus showed age-related decreases in response to motor inhibition, consistent with prolonged SSRT in older individuals. In contrast, the right posterior hippocampus showed age-related increases in response to a subjectively estimated likelihood of the stop signal or p(Stop). As higher estimates of p(Stop) prolong go trial reaction time—a manifestation of proactive control—this preparatory activity may have contributed to a compensatory process to maintain the sequential effect in older individuals.

### 4.3. Structural and Functional Asymmetry of the Hippocampus

Previous studies have reported hippocampal volumetric asymmetry in both healthy and pathological aging, with the right hippocampus having greater volume than the left [82]. Such asymmetry might be related to a faster age-related volume reduction rate of the left than right hippocampus [83]. Hippocampal asymmetry appeared to be most prominent in patients with AD, followed by those with MCI, and the least in healthy controls [84,85] (however, see [82]). In MCI, the right but not the left hippocampal volume was positively associated with performance in a spatial contextual cueing task, suggesting functional laterality of the hippocampus [5]. Indeed, the right but not left hippocampus in healthy older adults showed higher activation during high confidence hits versus misses during memory retrieval [86]. Our result of age-related changes in the right hippocampus is hence consistent with previous research.

## 5. Limitations, Other Considerations, and Conclusions

A number of limitations need to be considered for the current study. First, this is a cross-sectional study, and the current findings do not provide information as to how hippocampal function and dysfunction may evolve through the development of MCI and early-stage AD. Likewise, more studies are needed to understand whether or how these neural markers may predict the development of dementia. Second, the hippocampus partakes in a wide variety of cognitive processes. With data collected from a single behavioral task, we are not able to evaluate whether age is associated with similar extents of impairment across the cognitive domains as a result of hippocampal dysfunction. Third, likely as a result of the small sample size and relative paucity of older individuals, we were not able to establish a statistically significant mediating effect of hippocampal activation on age-related changes in SSRT or of hippocampal GMV on age-related changes in activation to the proactive control. The relationship between age-related changes in hippocampal volume and activities remains to be investigated. Fourth, besides the sequential effect, proactive inhibition can also be quantified as a frequentist measure [87]. As shown in our earlier work, this and the Bayesian measure were highly correlated [88] and demonstrated similar age-related changes (Figure A2). Finally, women and men may demonstrate differences in age-related hippocampal processes, and a larger sample size with balanced age distribution would be needed to address this issue.

In conclusion, we showed that age is associated with diminished capacity in reactive response inhibition, and reduced activation of the right anterior hippocampus contributes to this age-related deficit. Proactive control, as quantified by the sequential effect in the stop signal task, appears to be preserved in older adults, which may at least in part be accounted for by higher age-related activation of the right posterior hippocampus. The findings extend the literature by implicating the hippocampus in age-related changes in cognitive control.

## Figures and Tables

**Figure 1 brainsci-10-01013-f001:**
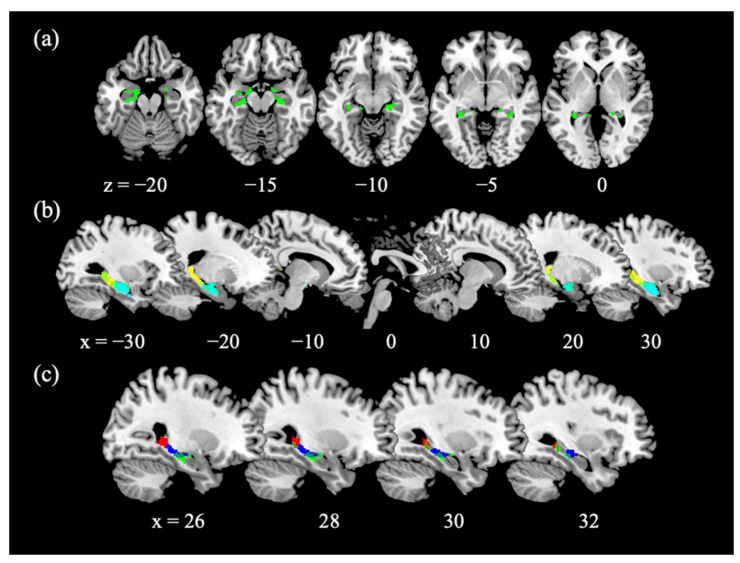
GMV of the hippocampus. (**a**) Age-related decreases in the GMV of bilateral anterior and posterior hippocampi; (**b**) Overlap with anterior hippocampus mask (cyan) and posterior hippocampus mask (yellow) divided at MNI coordinate y= −22, as defined in Zeidman and Maguire [30]; (**c**) Overlap with the functional cluster in the anterior hippocampus identified from a regression of SS > GS with age (blue; Figure 2a) and with the functional cluster in the posterior hippocampus identified from a regression of p(Stop) > 0 with age (red; Figure 2a).

**Figure 2 brainsci-10-01013-f002:**
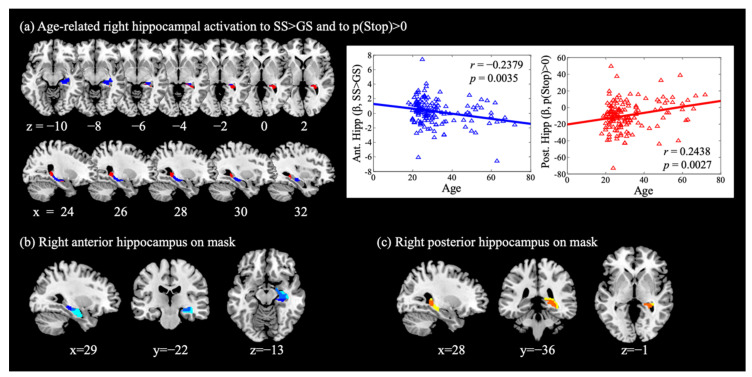
Hippocampal activations during cognitive control. (**a**) Right anterior hippocampus (blue) showed activation during SS > GS in negative correlation with age; right posterior hippocampus (red) showed activation to p(Stop) > 0 in positive correlation with age. These clusters were overlaid on (**b**) the right anterior hippocampus mask (MNI coordinate anterior to y = −22, cyan, as defined in Zeidman and Maguire [30]; overlap shown in cobalt), and on (**c**) the right posterior hippocampus mask (yellow) with the overlap shown in orange.

**Figure 3 brainsci-10-01013-f003:**
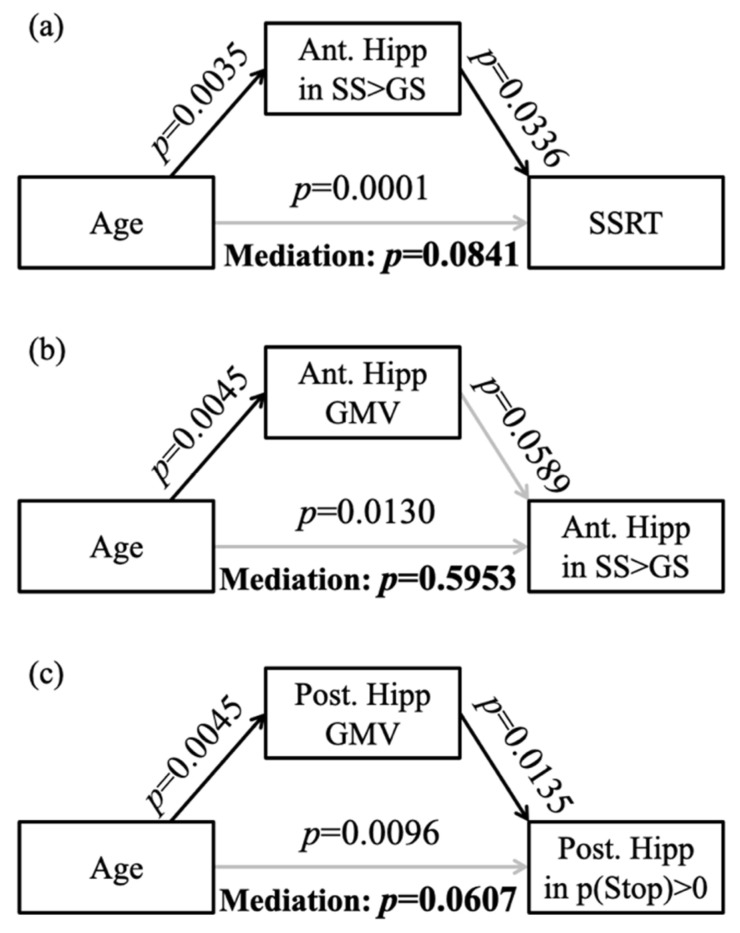
Mediation analysis between (**a**) age and SSRT with the right anterior hippocampal activation to SS > GS being the mediator, (**b**) age and the right anterior hippocampal activation to SS > GS with the GMV of right anterior hippocampus being the mediator, and (**c**) age and the right posterior hippocampal activation to p(Stop) with GMV of right posterior hippocampus being the mediator. Ant. Hipp: anterior hippocampus; Post. Hipp: posterior hippocampus; gray line: not significant result.

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
