# Peer review of "Age-Related Structural and Functional Changes of the Hippocampus and the Relationship with Inhibitory Control"

_brainsci, 2020, doi:10.3390/brainsci10121013_

Round 1

Reviewer 1 Report

Reviewer Report.

The paper “Age-related structural and functional changes of the hippocampus and the relationship with inhibitory control” by Hu and Li presents applications of two available statistical methods in the SST literature to explore two potential markers of cognitive aging. While the empirical side of the paper sounds solid and strong, its statistical side and writing style has some problems that need to be addressed and clarified. The authors are encouraged to address these issues in the revision and resubmit the paper.

Abstract.

  • The term sequential effect as the index for proactive inhibition is vague inline 17. The explanations after it make vaguer and the reader loses it. It is better to make it more concise and on point.

Introduction.

  • In lines, 70-76 the authors' outline and structure of the results and the paper is insufficient and self-interrupting. It is better this paragraph to be rewritten. It is better with one sentence to indicate the reader what will come in the paper: Example: The outline of our results is as follows: First, …; Second; ….; Finally, ……

Materials and Methods.

  • Line 85: Mention it clearly that this is standard Stop Signal Task(SST). There are several types of SST in the literature.
  • Lines 90-95:  Report the total number of SST trials per participant.
  • Lines 112-128: Clearly mention what is proactive inhibition index. Here, it is better inline 125 the authors mention something like this:

The proactive inhibition index was formulated via sequential effect as :

     Seq.Effect=Corr(p(stop),goRT)                             (4)

  • Lines 198-199: Define the terms  i1,i2,i3, e1,e2,e3 in the equation (1)-(3).

Discussion.

  • In one paragraph address the following question:

What are the advantages and disadvantages of choosing a frequentist based calculation method for reactive inhibition(SSRT) as in reference [36] and a Bayesian-based calculation based for proactive inhibition(Seq.Effect) as in references[30,37,38] . Why not either both Bayesian-based or both frequentist based?

  • In one paragraph address the following question:

Some literature considers proactive inhibition as a frequentist index quantified by the difference between trial type related goRTs. One such paper from the Brain Science Journal is cited below.  Why did not the authors consider both reactive inhibition(SSRT) and proactive inhibition( from the same type of measurement of time in milliseconds?   What would be the advantages and disadvantages of such consideration?

  • Soltanifar M, Knight K, Dupuis A, Schachar R, Escobar M. A Time Series-Based Point Estimation of Stop Signal Reaction Times: More Evidence on the Role of Reactive Inhibition-Proactive Inhibition Interplay on the SSRT Estimations. Brain Sciences. 2020; 10(9):598.

Before the Reference Section.

  • Add the list of abbreviations and their expanded expressions used in the paper in alphabetical order in two columns for the readers’ referrals.

Reviewer 2 Report

Hu and Li conducted a study on the cross-sectional associations between inhibitory control and hippocampal structure/function in a sample of adults aged 18 to 72 years old. Results suggested the expected age-related decrease in grey matter volume, which itself was unrelated to behavioral performance. Differences were observed between the right-lateralized anterior and posterior sections of the hippocampus in relation to age and behavioral measures. However, mediation analyses were not significant, leaving uncertain the link between age-related structural/functional changes and behavioral correlates of inhibitory control.

I commend the authors for creating a well-written and clear manuscript; it was a pleasure to read. The sample is a relatively good size for a neuroimaging study, and the methodology is sound. The choice of focusing on voxels overlapping between structure and functional effects in the mediation analyses is very nice.

The impact of this manuscript could be strengthened by some minor changes, as suggested below.

1) Is there any human literature on the role of the hippocampus in inhibitory/cognitive control that can be included in the introduction? The discussion on animal work showing a link between the hippocampus and control is good, but this argument could be strengthened by providing some work in humans (as you do in the second paragraph of the discussion).

2) More justification is needed regarding why inhibitory control matters in relation to MCI/dementia or aging. It's clear how hippocampal atrophy is relevant, but what makes inhibitory control in particular interesting to the reader? I think it's an interesting topic, and as a clinical psychologist with experience in neuropsychological assessment of MCI/dementia I understand why it could be relevant, and with some nuancing I think the authors can make their case more strongly in the introduction.

3a) Does age covary with sex in this sample? I'm wondering if there are sex-related differences in the age range, which may confound the observed age-related brain/behavior effects. Overlaid histograms of the age distributions for men and women would be helpful to visualize the sample.

3b) The authors state in the Limitations section that "a larger sample size with balanced age distribution" (line 339) is needed to address sex differences, which itself is an interesting question with public health significance. How unbalanced is the age distribution? Given that the sample is roughly evenly split between men and women, did the authors consider testing for sex interactions, or did the unbalanced nature of the sample preclude this?

4) Displaying the mediation analyses in a Table is fine, but readers may be more acquainted with visualizing mediation analyses in figures. The authors could consider replacing Table 1 with a figure or including an additional complementary figure.

5) Why do the primary GMV analyses use total (bilateral) volume whereas the primary BOLD analyses use right hemisphere activation? I see that GMV of left and right are presented in Table S1, but explicit justification should be made for this difference (total vs. left/right) in the analyses, or the same approach should be adopted across GMV and BOLD analyses to maintain a sense of consistency.

6a) Where are the results for the left hippocampus activation? As the manuscript is now, I don't know if only the right hippocampus BOLD signal was analyzed but not the left, or if the left hemisphere results were not significant and thus not reported. Even if they were not significant, in my view it's important to report them.

6b) I do not see any discussion on the hemisphere effects for the activation analyses (significant in the right but presumably not significant in the left). Was this expected?

7) Did the statistical models adjust for sex as covariate?

8) Some researchers may disagree, but I think the phrases "trend-level significance" (line 249) and "marginally" significant should be avoided. With the presumed alpha = 0.05 adopted in this report, the mediation effect is thus not significant per that cutoff. It is very possible that an effect is detectable with a larger sample, but the authors may reconsider this phrase.

9) I don't see how the lengthy discussion of fear extinction is very relevant.

10) The discussion could be improved by more emphasis on interpreting the current findings in terms of literature at-large related to aging/inhibitory control, how the findings may impact our understanding of diseases like MCI/dementia in relation to problems with impulsivity, disinhibition of inappropriate behaviors, etc.

All in all this is a well written manuscript on a well executed study that has potential relevance to a significant public health issue.

Round 2

Reviewer 2 Report

The authors have done a nice job of adequately addressing the comments and revising the manuscript. The additions and edits have strengthened the manuscript, and I can now recommend the acceptance of this manuscript.